# Clearance of Biodegradable Polymer and Polyethylene Films from the Rumens of Holstein Bull Calves

**DOI:** 10.3390/ani13050928

**Published:** 2023-03-03

**Authors:** Hailey Galyon, Samuel Vibostok, Jane Duncan, Gonzalo Ferreira, Abby Whittington, Kirk Havens, Jason McDevitt, Rebecca Cockrum

**Affiliations:** 1School of Animal Sciences, Virginia Tech, Blacksburg, VA 24061, USA; 2Department of Macromolecular Science and Engineering, Virginia Tech, Blacksburg, VA 24061, USA; 3Department of Chemical Engineering, Virginia Tech, Blacksburg, VA 24061, USA; 4Department of Materials Science and Engineering, Virginia Tech, Blacksburg, VA 24061, USA; 5Center for Coastal Resources Management, Virginia Institute of Marine Science, Gloucester Point, VA 23062, USA; 6William & Mary Research Institute, William & Mary, Williamsburg, VA 23187, USA

**Keywords:** biodegradable, polyhydroxyalkanoate, poly(butylene succinate-*co*-adipate), rumen, plastic impaction

## Abstract

**Simple Summary:**

Domestic and non-domestic ruminants indiscriminately ingest plastics, such as hay netting, typically developed from polyethylene. Due to their indigestible nature, plastics accumulate in the rumen when ingested and animal health and production problems arise. Therefore, it is necessary to find biodegradable materials that will break down in the rumen when consumed to replace polyethylene. In pursuit of this, we fed twelve Holstein bull calves capsules containing a biodegradable polymer, polyethylene, or nothing for 30 d and evaluated animal health. At the end of the 30 d, animals were euthanized to evaluate the rumen for potential damage and to characterize polymer residues in rumen contents. No calves presented any signs related to the building up of polymer in the rumen and animal health was not affected by any treatment. Calves that received polyethylene retained 7% of the total dosed material, while calves that received the biodegradable polymer retained less than 1%. The biodegradable residues were fragmented to 10% of their original size. Based on our results, we suggest that agricultural plastics developed from biodegradable polymers may be a potential alternative to conventional plastics in the case of animal ingestion due to their improved ruminal degradation and decreased accumulation in the rumen compared to conventional polyethylene.

**Abstract:**

Due to the occurrence of plastic impaction in ruminants and its deleterious effects on health and production, it is necessary to determine the suitability of biodegradable polymers to replace polyethylene-based agricultural plastics, such as hay netting. The objectives of this study were to evaluate the clearance of a polyhydroxyalkanoate (PHA) and poly(butylene succinate-*co*-adipate) (PBSA) melt-blend polymer from the rumen when fed to cattle and subsequent animal health. Twelve Holstein bull calves were dosed with an encapsulated 13.6 g of PBSA:PHA (Blend), 13.6 g of low-density polyethylene (LDPE), or four empty gelatin capsules (Control) for 30 d. The feed intake, body weight, and body temperature were evaluated, and hemograms were run on d 0 and d 30. On d 31, calves were euthanized to evaluate gross rumen measurements and pathology, papillae length, and polymer residues in rumen contents. No calves presented any signs related to plastic impaction. The feed intake; body weight; rectal temperature; hematological parameters; gross rumen measurements and pathology; and rumen pH and temperature were not affected by treatments. Calves dosed with LDPE had 27 g of undegraded polymer retained in the rumen while Blend calves had only 2 g of fragmented polymers that were 10% of their original size. Agricultural plastics developed from PBSA:PHA may be a suitable alternative to LDPE-based products in the case of animal ingestion and may reduce the incidence of plastic impaction.

## 1. Introduction

Plastic impaction is detrimental to ruminant health and should be a concern for the dairy industry as it unfavorably impacts the rumen environment and animal productivity. Though no surveys have yet evaluated occurrences in the United States, it is estimated that 20% of cattle are affected by plastic impaction, according to abattoir studies in the East [1,2]. Plastic impaction occurs when large amounts of indigestible plastic materials are indiscriminately ingested by ruminants in either a single meal or over the lifetime of the animal. These materials may be introduced into cattle rations by chopping forages with intact netting. In this condition, it is estimated that 0.07% of the diet contains polyethylene-based plastics [3]. When these materials enter the reticulorumen, they aggregate together to form large ball-like masses that press against rumen epithelium during rumination [3,4,5,6,7]. With a long-term presence, plastics reduce rumen functionality and may impact feed efficiency. This is through decreasing ruminal movement and the methylene blue reduction time of rumen fluid, an indirect measure of microbial fermentation [8], and inducing pathological alterations to the rumen epithelium [2,6,9]. To reduce the incidence of plastic impaction, it is crucial to identify biodegradable alternatives to polyethylene that will safely degrade within the rumen environment with no harmful effects on animal health. Potential alternative plastic materials include polyhydroxyalkanoates (PHAs) and other biodegradable polymers such as poly(butylene succinate-*co*-adipate) (PBSA).

We previously evaluated the in vitro ruminal degradation of PHA, PBSA, PBSA:PHA melt-blend, and low-density polyethylene (LDPE) polymers in rumen fluid for 10 days. This study demonstrated the potential for biodegradable polymers to begin the degradation process within 24 h in rumen fluid [10]. We further evaluated the in situ ruminal degradation of PHA, PBSA, PBSA:PHA, and LDPE films in dairy cattle for 150 d [11]. The results of this experiment confirmed that polyethylene materials do not degrade within the rumen and that a PBSA:PHA blend may be the best material to replace LDPE for net wrap applications. Although mass loss did not occur until 120 d, the PBSA:PHA blend began to fragment to 29% of its original size by just 24 h of inclusion in the rumen [11]. Thus, this polymer blend may fragment enough to pass through the reticulo-omasal orifice and through the rest of the digestive tract, decreasing the incidence of plastic accumulation within the rumen compared to polyethylene-based products. Before the development of plastic net wraps from a PBSA:PHA material can be considered, in vivo studies that evaluate the clearance of PBSA:PHA from the reticulorumen after feeding and subsequent influences on animal health are necessary.

The objectives of this study were to (1) evaluate the clearance of a PBSA:PHA melt-blend polymer from the rumen compared to LDPE and (2) evaluate animal health when these materials are fed to cattle. We hypothesized that, unlike LDPE-fed cattle, PBSA:PHA would fragment in the rumen and pass through the reticulorumen and that health parameters would not differ from healthy control cattle.

## 2. Materials and Methods

### 2.1. Treatment Preparation

A 10% PHA and 90% PBSA polymer blend (*w*/*w*) was developed by melt-blending and extruding a proprietary PHA-based polymer (Mirel P1004, Metabolix, Inc., Woburn, MA, USA) and PBSA (Bionolle 3001MD, Showa Denko America, Inc., New York, NY, USA) into a nurdle formation using a pilot-scale extruder at Alterra Plastics (Clifton, NJ, USA). Low-density polyethylene nurdles were purchased from Sigma-Aldrich, Inc. (St. Louis, MO, USA). Films were produced from polymer nurdles using a melt press (#SP210C-X351220, PHI Hydraulics, Inc., City of Industry, CA, USA). Nurdles were placed between two non-stick Kapton sheets on platens set to 174 °C and incremental pressure to 340 kg. The bottom platen was raised near the top platen, and the polymer nurdles were given roughly 30 s to soften before the bottom platen was raised entirely. The polymer nurdles were melted for approximately 30 s to form circular films 0.1 mm thick. Circular films were then run through a cross-cutting paper shredder (#ITSH-555, Innovative Technology Americas, Inc., Pewaukee, WI, USA) with 4 mm blades to produce PBSA:PHA and LDPE film strips that were 50.4 ± 19.1 mm long. 

To prepare daily boluses for treatments, 3.4 g of PBSA:PHA or LDPE films was packed in 7.5 mL Torpac gelatin capsules (Fairfield, NJ, USA). Three daily treatments were utilized in this study: 4 empty gelatin capsules (Control), 13.6 g of PBSA:PHA in 4 gelatin capsules (Blend), or 13.6 g of LDPE in 4 gelatin capsules (LDPE). 

### 2.2. Calf Health and Management

The Virginia Polytechnic Institute and State University Animal Care and Use Committee approved all animal procedures (IACUC #19-265). Calves were chosen as the subject for this study due to their heightened sensitivity to toxicities compared to fully mature cows. In this 6-week study, 12 weaned Holstein bull calves were purchased from a single farm (Floyd, VA, USA) and transported to Virginia Tech (Blacksburg, VA, USA). Upon arrival, calves were evenly grouped into 4 blocks by age (62 ± 9 d) and initial body weight (74.9 ± 8.0 kg). Calves were individually housed in tie-stall pens (122 cm × 183 cm) in temperature- (20 °C) and light-controlled rooms (12 h of light) for the duration of the study. Pens were placed on top of rubber mats (122 cm × 183 cm) and pine shavings were provided for bedding. Calves had visual and auditory contact with each other, as well as nose-to-nose contact between adjacent stalls. 

Calves were acclimated for 14 to 15 d before the start of the treatment period depending on the start of treatment administration as described below. Upon arrival, calves were offered 2.3 kg of calf starter (Producer’s Pride, Brentwood, TN, USA), ad libitum mixed-grass hay, and ad libitum water. According to the manufacturer’s specifications, the calf starter contained 16.2% crude protein, 1.6% crude fat, 13.0% crude fiber, and 17.0% acid detergent fiber. On a dry matter basis, the mixed-grass hay contained 12.7% crude protein, 61.6% neutral detergent fiber, 36.3% acid detergent fiber, and 3.8% lignin. Calves were acclimated to 2.7 kg of starter by d 0 and continued to receive ad libitum mixed-grass hay and water. Feeding occurred once daily at 07:00 and refusals for starter and hay were weighed separately the next morning before feeding. 

Calves were randomly assigned to treatments as a randomized complete block design such that within each of the four blocks one calf each was assigned to Control, Blend, or LDPE treatment. Treatment administration occurred directly after feeding via a plastic bolus gun from d 0 to d 30. Two blocks were randomly selected to start one day, and the remaining two blocks started the next to ensure treatments were administrated for 30 d before tissue collection on two consecutive dates. The treatment period of 30 d was chosen such that, if polymer did not escape the rumen, 408 g (approximately 0.5% of the calf body weight) would be retained in the rumen. It was hypothesized that this level of accumulation would cause histopathological damage as seen by Otsyina and colleagues [6].

Calf health was assessed twice daily, at 07:00 with feeding and again at 15:00. Rectal temperature, fecal score, obvious treatment presence in fecal matter, bloat occurrence, eye score, nasal score, and activity were evaluated. According to the University of Wisconsin Madison School of Veterinary Medicine’s Calf Health Scoring Chart, fecal score was evaluated with a 5-point scale: 1 = watery, 2 = loose, 3 = semi-formed and pasty, 4 = hard pellets, and 5 = normal. Eye score was evaluated with a 4-point scale: 1 = heavy ocular discharge, 2 = moderate bilateral discharge, 3 = limited discharge, and 4 = no discharge. Nasal score was evaluated with a 4-point scale: 1 = copious amounts of mucus discharge, 2 = bilateral cloudy mucus discharge, 3 = small amount of unilateral discharge, and 4 = no discharge. It was noted if a calf exhibited a fecal, nasal, or eye score equal to or below 2, lethargy with depressed ears, a cough or elevated respiration rate, a rectal temp above 40.0 °C, or a reduced feed or water intake. A veterinarian was sought if any of these signs presented for more than a day or if two or more of these symptoms occurred simultaneously. Every week, calves were weighed by walking onto scales, and average rectal temperature, daily grain intake, and daily hay intake were determined. 

### 2.3. Hematological Evaluation

Whole blood samples were collected from animals via the venipuncture of the jugular vein into 7 mL vacutainers containing potassium ethylenediaminetetraacetic acid (_3_EDTA) on d 0 and d 30 of bolus administration. Within an hour of collection, approximately 1 mL of blood mixed with EDTA was taken up by a syringe for determination of erythrocyte sedimentation rate (ESR) using the Wintrobe tube method to determine potential inflammatory activity. The blood was deposited into perpendicular Wintrobe tubes to the zero mark. After 24 h, the fall of red-packed cells was measured in millimeters. 

The remaining blood samples were delivered to the Virginia Tech Animal Laboratory Services (VITALS) within 2 h of collection for the determination of red blood cells (RBCs), hemoglobin (HB), hematocrit, mean corpuscular volume (MCV), mean corpuscular hemoglobin concentration (MCHC), red cell distribution width (RDW), reticulocytes, white blood cells (WBC), neutrophils, monocytes, eosinophils, basophils, platelets, and mean platelet volume (MPV) via a Sysmex XN-1000 (Sysmex America, Inc., Lincolnshire, IL, USA).

### 2.4. Tissue Collection and Gross Rumen Measurements

Calves were slaughtered via captive-bolt stunning followed by exsanguination on d 31 between 08:00 and 11:00, at least 2 h after feed was removed. The entire gastrointestinal tract was ligated and removed from the carcass. The full forestomach was removed, weighed in full, and then segmented at the reticulo-omasal orifice. The reticulorumen was then weighed in full before being opened to measure the pH and temperature of the rumen contents immediately. A small sample of total contents was strained with cheesecloth by hand to retrieve approximately 10 mL of rumen fluid to determine its methylene blue reduction time (MBRT). The MBRT was determined as the time for the blue coloration to disappear from the rumen fluid after 0.5 mL of 0.04% methylene blue solution was added [12]. The total rumen contents were then thoroughly mixed by hand and a small sample was collected, approximately 200 g. The total weight of the sample was determined, and then the rumen solids were strained through two layers of cheesecloth and reweighed. Solids were flash-frozen until further analysis to determine the amount and size of polymer residues present. All solids were evacuated from the reticulorumen and then subjectively scored using a 5-point scale: 0 = no polymer present, 1 = small amount of degraded polymer present, 2 = degraded polymer present throughout contents, 3 = small amount of intact polymer present, and 4 = intact polymer present throughout contents.

The empty reticulorumen was weighed to determine the total rumen contents and then rinsed to remove any remaining particles loosely attached. The reticulorumen was “butterfly-cut” for a full visual of the epithelium. The epithelium was grossly evaluated in the cranial dorsal, caudal dorsal, cranial ventral, and caudal dorsal regions using a 6-point scoring system following Jonsson et al. [13]: 0 = no evidence of any damage, 1 = palm-size or smaller areas bare of papillae, 2 = large areas bare of papillae, 3 = areas of scarring, 4 = red/bloody areas, and 5 = areas of parakeratosis. Rumen epithelium samples for papillae measurements were obtained from the cranial dorsal and caudal ventral regions using a scalpel and a 3 cm × 10 cm stencil. Epithelium samples were stapled to wooden tongue depressors and immediately placed in 10% formalin to fix for 48 h. 

### 2.5. Rumen Histological Measurement

Epithelium sections of 0.5 cm were cut from fixed tissue samples, placed in cassettes in 10% formalin, and sent to VITALS for routine processing. This included embedding in paraffin, sectioning 5 μm thick sections and mounting them onto three positively charged microscope slides, and hematoxylin and eosin staining following standard procedure. Digital images of slides were taken at 4x using an Olympus BX43 microscope (Olympus Corporation of the Americas, Center Valley, PA, USA) fitted with a Retiga R6 camera (QImaging Corporation, Surrey, BC, Canada). Images were viewed in ImageJ (U.S. National Institutes of Health, Bethesda, MD, USA) for measurement of papillae length using the line tool. Three random papillae from each slide were selected and measured.

### 2.6. Polymer Residues

Frozen rumen solids samples were thawed to room temperature and then dried in a forced-air oven at 55 °C for 48 h. The total dried sample weight was determined and polymer particles were manually removed with forceps. Residual polymers were weighed in total and then three random particles were selected, and their length was measured with digital calipers. The total residual polymer present within the rumen was estimated using the following:(1)$$\mathrm{Accumulation}\;\left(g\right)=contents\;\left(g\right)\times solids\;\left(\%\right)\times DM\;\left(\%\right)\times polymer\;\left(\%\right)$$
where *contents* indicates the total weight of all rumen contents, *solids* indicates the ratio of solids to total weight of rumen contents, *DM* indicates the dry matter content of solids, and *polymer* indicates the ratio of polymer to dry solids weight. 

### 2.7. Statistical Analysis

Unless otherwise mentioned, all data were analyzed using the MIXED procedure and least-squared differences were adjusted by Tukey’s method in SAS 9.4 (SAS Institute, Cary, NC, USA). Following a Box–Cox transformation, the average daily grain intake was transformed by the power of 5, and the average daily hay intake was transformed by taking the square root to improve the normality of the data for statistical analysis. Grain intake, hay intake, body weight, and rectal temperature were analyzed with the fixed effects of treatment, week, the interaction, and the random effect of block. Weekly observations were the repeated measures and animals were the subjects. The autoregressive covariance structure was used according to the AIC criterion. Hematological parameters were analyzed with the fixed effects of treatment, sampling day, the interaction, and the random effect of block. The sampling day was the repeated measure and animals were the subjects. The autoregressive covariance structure was used according to the AIC criterion. Subjective scoring parameters of rumen contents and gross pathology of epithelium regions were analyzed using PROC FREQ and the fixed effect of treatment. Papillae length was analyzed with the fixed effects of treatment, region, the interaction, and the random effects of block and slaughter date. Forestomach weights, MBRT, rumen pH and temperature, estimated polymer accumulation, and polymer length were analyzed with the fixed effect of treatment and the random effects of block and slaughter date. Statistical significance was determined at *p* < 0.05 and statistical tendency was determined at *p* < 0.10. 

## 3. Results

### 3.1. Animal Growth and Health

Neither average daily grain nor hay intake were affected by the interaction of treatment and sampling day, nor treatment by itself (Figure 1). However, both daily feed intakes increased with time. On an as-fed basis, calves ate 1.77 kg of grain on average during the second acclimation week. Average daily grain intake increased by 16% by d 0 (*p* < 0.01) and this percent increase was maintained to d 7 (*p* < 0.01). The increase in grain intake slowed to an 11% increase from d 7 to d 14 at 2.66 kg (*p* < 0.01), after which the average daily grain intake was maintained through d 28. Meanwhile, the average daily hay intake increased throughout the study at a slower rate. Calves ate very little hay during the second acclimation week at only 0.05 kg. This increased threefold by d 0 (*p* < 0.01) and then nearly twofold by d 21 (*p* < 0.01). By d 28, the average daily hay intake reached 0.50 kg. The increased feed intake over time was accompanied by increased body weight. The body weight of bull calves was also not affected by the interaction or treatment by itself. The body weight increased linearly from week to week by approximately 5% (*p* < 0.01; Figure 2). At d 7, calves were 79.01 kg, and by d 28, they increased to 104.55 kg with an average daily gain of 0.73 kg/d.

Regarding calf health, rectal temperature was not affected by any of the fixed effects and averaged 38.1 °C. Calves did not demonstrate bloat, scours, absence of fecal matter, reduced feed intake, reduced body weight gain, or abnormal behaviors such as kicking at the stomach and stomping that would have indicated any gastrointestinal blockage due to polymer build-up in the tract. However, calves did develop signs of bovine respiratory disease shortly after arriving at Virginia Tech. All calves were blanket-treated with tulathromycin (IncrexxaTM, Elanco, Greenfield, IN, USA) and an intranasal bovine rhinotracheitis-parainfluenza 3-respiratory syncytial virus vaccine (Inforce 3^®^, Zoetis, Parsippany, NJ, USA) by the attending veterinarian. By d 0, signs of illness subsided and calves were deemed healthy to begin the study.

### 3.2. Hematological Parameters

Hematological parameters were unaltered by the administration of polymer treatments (Table 1). When comparing blood parameters from d 0 to d 30 within treatment groups, only WBC, neutrophil count, and platelet count seemed to be affected. The white blood cell count tended to decrease by 35% in LDPE calves (*p* = 0.08) but did not change for the other two groups. Neutrophil counts tended to decrease by 76% in Control calves (*p* = 0.07) and nearly halved in LDPE calves (*p* = 0.03) but did not change for Blend calves. The platelet count decreased by 30% for Control calves (*p* = 0.01) and decreased by 40% for LDPE calves (*p* < 0.01) but did not change for Blend calves. 

### 3.3. Rumen Measurements

Rumen measurements are displayed in Table 2. Gross rumen measurements of total forestomach weight, reticulorumen weight, empty reticulorumen weight, and total rumen contents did not differ among the treatment groups. Rumen temperature and pH also did not differ among treatments. However, there was a tendency for MBRT to be affected by treatment. Blend calves tended to have reduced MBRT compared to LDPE calves by 46 s (*p* = 0.06).

Gross evaluation of epithelium in the cranial dorsal, caudal dorsal, cranial ventral, and caudal ventral regions of the reticulorumen epithelium did not reveal any significant differences between treatment groups (*p* = 0.34). In Control and LDPE calves, no region had evidence of any damage as indicated by a score of 0. One calf in the Blend group demonstrated a very small patch of bare papillae in both dorsal regions of the epithelium. However, these areas were not paired with any scarring, ulcerations, inflammation, or any other indicators of severe damage. 

### 3.4. Papillae Measurements

A treatment and epithelium region interaction existed for the lengths of papillae (*p* < 0.01; Figure 3). Within samples taken from the caudal ventral region of the reticulorumen, papillae of Blend calves were 4.24 mm and 50% longer than those of Control calves (*p* = 0.04) but did not differ from those of LDPE calves at 3.56 mm. Papillae from LDPE calves also did not differ from those of Control calves, which were just 2.84 mm long. The papillae length of the cranial dorsal region was not affected by treatment and averaged 4.33 mm. However, papillae from the cranial dorsal region were significantly longer by 61% than those from the caudal ventral region in Control calves (*p* < 0.01). In gross evaluation of reticulorumen integrity, no apparent disruption of the epithelium was appreciated for any treatment or region. Neither atrophy, an obvious stunting of papillae, necrosis, the shredding of stratified epithelium, nor hyperplasia were detected. 

### 3.5. Polymer Residues

In subjective scoring of rumen contents for polymer residues, treatment had a significant influence (*p* < 0.01). All Control calves received a score of 0, indicating no polymer presence. Blend calves all received a score of 1 indicating a small amount of degraded polymer present. Half of the LDPE calves demonstrated a score of 3 and the other half demonstrated a score of 4, indicating small and large amounts of intact polymer present within rumen contents, respectively. Polymer residues within rumen contents presented as individual film strips or shards. No apparent aggregation of polymer residues was evident with any treatment group.

This trend was reflected in the estimated polymer accumulation in the rumen and the length of residual polymers. Calves that received LDPE had significantly more polymer residues entrapped in the rumen compared to Blend calves (27.42 g vs. 1.72 g, *p* < 0.01; Figure 4A). Calves that received LDPE polymers retained approximately two days’ worth of polymer (7% of the total 408 g of polymer dosed), while Blend calves retained only 13% of a single day’s dosage (0.4% of the total 408 g dosed). Lengths of polymer residues were also different between treatment groups (Figure 4B). Polymers retained in the rumens of LDPE calves were 49.51 mm long, which was 10 times the length of Blend residues at just 4.81 mm long (*p* < 0.01).

## 4. Discussion

To our knowledge, this is the first study to specifically evaluate if a PBSA and PHA biodegradable polymer blend would degrade and pass from the reticulorumen of dairy cattle if ingested compared to a polyethylene control. We hypothesized that LDPE, given its indigestible nature, would accumulate in the rumen and unfavorably impact animal health, while PBSA:PHA would fragment and not impact calf health. However, our findings demonstrated that neither a PBSA:PHA polymer melt-blend nor traditional LDPE influenced the health or growth of Holstein bull calves at approximately 0.5% inclusion of the diet, as fed, for 30 d. Due to increased rumen filling by the accumulation of plastics and other indigestible foreign materials in the rumen and subsequent activation of stretch receptors in rumen epithelium, satiety centers in the hypothalamus are stimulated and create a sense of fullness by the animal [14]. A decreased feed intake would thereby reduce available energy for the animal and weight gain would theoretically decrease. Though limited studies evaluate plastic impaction in cattle in controlled environments, studies that evaluated animals diagnosed with plastic impaction at abattoirs demonstrated these characteristics. Animals with plastic foreign bodies within the rumen consumed less food and were subsequently 13% lighter than healthy animals [7]. Animals similarly demonstrated poor body condition scores highly correlated to increased amounts of plastic materials trapped in the rumen [15,16,17]. However, these studies were more observatory and evaluated animals after accumulation occurred over a productive lifetime. The depth at which plastic accumulation influences these parameters and a threshold value before the feed intake and body weight decrease in cattle cannot be estimated at this time. The calves utilized in this study were only 2 months old and presumably not exposed to plastic materials before this study. Only 27.42 g and 1.72 g of LDPE and PBSA:PHA were estimated to be maintained in the rumen after 30 d of daily ingestion. This is equivalent to 0.3% and 0.02% of total rumen contents. Likely, polymer accumulation in this study was not enough to stimulate rumen filling and satiety to reduce the feed intake and, subsequently, reduce growth. 

Previous observational and controlled studies of plastic impaction in ruminants demonstrated alterations in hematological parameters compared to healthy animals. Observational studies found that the WBCs [8,18], RBCs, and Hb [19] were reduced in animals with plastic impaction while the MCV and MCHC were elevated [19]. A controlled study in which sheep were implanted with plastic via rumenotomy for 42 d found, at just 129 g of plastic, or 0.5% body weight, sheep had reduced WBCs, RBCs, and HB and increased MCV. The MCHC was not elevated until 258 g of plastic implantation, and platelet counts were increased in animals implanted with 387 g [20]. Our study found the WBC to change in LDPE calves with a 35% reduction rather than an increase as previously seen. 

All our calves exhibited signs of bovine respiratory disease during the acclimation period. Though calves were treated and deemed healthy to begin the experiment by d 0, The WBC count could have still been elevated from the disease. Initial WBC counts for LDPE calves were the greatest of the three treatments and slightly elevated compared to a reference value of 8 wk old calves [21]. However, values still fell into the accepted range of 2.71 to 17.76 × 10^3^ cells/uL as provided by VITALS. It is likely that the WBC count reduction after 30 d was due to recovery from the disease rather than LDPE administration as the Control and Blend groups also had slight reductions despite not being significant. Neutrophil counts also decreased for Control and LDPE calves after 30 d. However, neutrophil counts remained within the normal range of 0.7 to 6.9 × 10^3^ cells/uL as given by VITALS. Again, the trend of decreasing over time is likely explained by the recovery of animals from bovine respiratory disease during the acclimation period rather than a treatment effect. This is especially so as Control animals displayed this trend. Platelet counts similarly decreased after 30 d for Control and LDPE calves. Platelets play a key role in the inflammatory response and counts increase during times of inflammation throughout the body [22]. Bovine respiratory disease is highly associated with the inflammation of the respiratory tract. Elevated levels of platelets in this study may be due to this inflammatory response. Platelet counts returned to normal values after 30 d, indicating the recovery of the animals and a lack of rumenitis due to polymer presence. The absence of a true treatment effect, namely LDPE, on changing histological parameters can be explained by the small accumulation of polymer within the rumen contents and the short duration of the study. 

The gross rumen development was not affected by treatment; the total forestomach weight, total reticulorumen weight, and empty reticulorumen weight were not different among treatments. The influence of plastic impaction on rumen growth has not yet been evaluated in any ruminant species. As previously described, most studies evaluating plastic impaction are post-impaction observations of mature animals when rumen growth and development have ceased. Because polymer accumulation was slight in this study and only for 30 d, it is unknown if polymer presence would influence rumen development in growing animals. Severe plastic impaction in young animals may lead to a reduced feed intake and therefore decreased available nutrients to support the growth of the gastrointestinal tract. However, it is doubtful that a severe accumulation of plastics would occur in young animals in standard production conditions in the United States. Total mixed rations contain approximately 0.07% chopped net wrap on an as-fed basis when hay bales are ground with intact netting [3]. Thus, accumulation may be slow enough that rumen development would be complete before a significant influence by LDPE-based materials. No studies to date have specifically evaluated this and further studies are needed to specifically evaluate the effect of plastic impaction on rumen growth in developing animals under normal feeding conditions. 

Similarly, internal rumen parameters did not differ among treatments. Previous observational studies indicate that animals with plastic foreign body impaction typically have an increased MBRT and rumen pH compared to healthy animals [8,17,23]. The methylene blue reduction time is an important parameter to evaluate as it is indicative of the fermentation redox potential and activity of microbes in rumen fluid. Healthy animals typically have an MBRT of 180 s or less, while animals with plastic impaction may have fourfold increases depending on the level of accumulation [8,23,24]. This increase in the MBRT is due to the decreased activity of microbes, likely due to the reduced rumen motility and reduced feed intake of animals with severe plastic impaction [16]. The average MBRTs of Control, Blend, and LDPE calves were all under 180 s, and thus it can be concluded that microbial fermentation was not particularly diminished. A potential explanation for the reduced MBRT in Blend calves compared to LDPE calves could be degradation products released from the PBSA:PHA melt-blend, more specifically the PHA component. Although we have yet to evaluate specific degradation products released by this polymer in rumen fluid, butyrate concentrations within rumen fluid may increase due to the release of 3-hydroxybutyrate from PHA biodegradation [25]. Depending on the extent of butyrate released into the rumen pool, total volatile fatty acid concentrations may increase and reduce the rumen redox potential [26]. More negative rumen redox potential is highly associated with improved microbial fermentation activity [27]. It may be through this mechanism that the MBRT is reduced in animals dosed with the PBSA:PHA melt-blend. However, if volatile fatty acid concentrations were increased in Blend animals, they were not sufficient enough to decrease pH as previously seen in feeding studies of poly(3-hydroxybutyrate) to aquatic species such as the European sea bass [28].

Further studies with more animals fed a PBSA:PHA melt-blend should evaluate its biodegradation products, its influence on the microbial ecology, and the redox potential of the rumen to sufficiently determine the long-term influences of this polymer on ruminal fermentation.

In gross evaluation of the epithelium, observational studies consistently found that animals with plastic foreign bodies present with rumenitis, hemorrhaging, erosion and sloughing of rumen mucosa, lesions, regions bare of papillae, and bent and stunted papillae [4,5,6,7]. These observations were explained by the aggregation of plastic materials forming ball-like masses, which press against the rumen epithelium during rumination. These observations were not evident in the present study with either Blend or LDPE treatments in any of the four regions of the reticulorumen. Polymer residues in this study did not aggregate together to form masses of material and polymers seemed to be equally distributed among the rumen contents. The lack of aggregation of the limited polymer residues likely explains the absence of pathological damage to the rumen epithelium and mucosa. To fully appreciate if a PBSA:PHA blend polymer would induce pathological damages to the rumen compared to a LDPE polymer control, feeding studies of a longer duration with netting prototypes developed from these materials need to be conducted. 

Though gross pathological changes were not observed, papillae length was influenced by polymer introduction to the rumens of Holstein bull calves. A previous study measured the lengths of papillae from the dorsal rumens of goats and ewes that presented plastic foreign bodies. Compared to healthy animals, papillae from affected animals were 60% shorter [7]. Additionally, papillae length was negatively correlated to the amount of foreign material present within the rumen. In the present study, papillae length within the cranial dorsal region of calves did not differ among those that received either a biodegradable polymer or LDPE compared to animals that did not receive any polymer. The drastic difference in accumulation between the previous study and this one likely explains this. Though the duration of accumulation in the referenced study is unknown, plastic accumulation was on average 2.75 kg [7]. This is 100 times more than what accumulated in our study over 30 d. Interestingly, in the caudal ventral region, Blend calves’ papillae were 50% longer than Control calves. It is possible that the degradation of the PHA component of the PBSA:PHA melt-blend introduces exogenous butyrate into the rumen. Butyrate is a vital modulator in the functional development of the rumen epithelium and is involved in cellular proliferation pathways [29,30]. As such, butyrate potentially released from this polymer may have stimulated epithelium development and increased papillae lengths. However, Blend calves’ papillae length in the caudal ventral region were not different from the papillae of LDPE calves. Repeated studies with more animals for a longer duration to accrue more polymer residues are necessary to further discern this relationship. 

Despite the apparent health of animals in the face of LDPE administration for 30 d, the characterization of residual polymers in rumen contents indicates PBSA:PHA may be a better material than LDPE for single-use agricultural plastics in the case of ingestion. Animals dosed with 13.6 g of LDPE daily had 16 times more polymer residues entrapped in the rumen than those dosed with 13.6 g of PBSA:PHA daily. This indicates that accumulation occurs at a much faster rate for LDPE than PBSA:PHA material. However, the accumulation of LDPE in the rumen was lower than expected. A study in which a total mixed ration containing 0.07% chopped plastic net wrap was fed to cattle for 7 months found that 47.2% of net wrap remains in the reticulorumen in ball-like masses [3]. Our study found that only 6.7% of administered LDPE was retained. Given that this polymer is indigestible, as seen in our previous in situ results [11], polymers could have escaped the rumen and been excreted or regurgitated during rumination. The net wrap feeding study also found that plastic particles that were retained in the rumen were more than 70 mm long in mature cattle [3] whereas the polymers utilized in this study were only 50 mm long. It is possible that some polymers were shorter than the average and escaped through the reticulo-omasal orifice. The lack of aggregation of polymer materials in the rumen also left polymers freely distributed among rumen contents. This could have increased the likelihood of ruminal escape. The absence of aggregation of the PBSA:PHA polymer may better be explained by the breakdown of the material as indicated by reduced fragment size. This reduced residual polymer size likely allowed PBSA:PHA to better escape the ruminal environment and thus decrease accumulation within the rumen.

It is important to note that this study utilized young Holstein calves, and characteristics of polymer degradation may not entirely reflect degradation in a mature cow. The ruminal microbiome is moldable within the first few months with the rumen being newly inoculated. It takes approximately 90 d for the rumen microbiome to be established and to stabilize [31]. As the ruminant matures, the microbiome continues to be inoculated with bacteria from the animal’s environment, thus providing more opportunities for polymer-degrading bacteria to be hosted. The degradation of PHA and PBSA may be improved in older animals due to increased microbial concentrations and the greater likelihood of PHA- and PBSA-degrading bacteria being present. Therefore, the passage of PBSA:PHA materials through the rumen may be improved in older animals that may consume these products.

## 5. Conclusions

Based on these preliminary findings, we suggest that feeding LDPE or PBSA:PHA polymer films at 13.6 g/d to Holstein bull calves for 30 d does not negatively impact animal health or development. Due to the undegradable nature of LDPE in the rumen, LDPE accumulated more than PBSA:PHA. Biodegradable PBSA:PHA degraded to smaller fragments that more readily escaped the reticulorumen. Based on these results, we hypothesize that PBSA:PHA may be a suitable material to replace polyethylene in the development of net wraps in the case of animal ingestion due to its reduced aggregation within the rumen. Future studies with greater animal numbers should evaluate the long-term inclusion of net wrap materials developed from PBSA:PHA in total mixed rations. Histological parameters; rumen development, pathology, and morphology; volatile fatty acid concentrations; polymer degradation products and their transference throughout the body; rumen fluid redox potential; the distribution of a net wrap throughout the entire gastrointestinal tract; and the net wrap passage rate should all specifically be determined.

## Figures and Tables

**Figure 1 animals-13-00928-f001:**
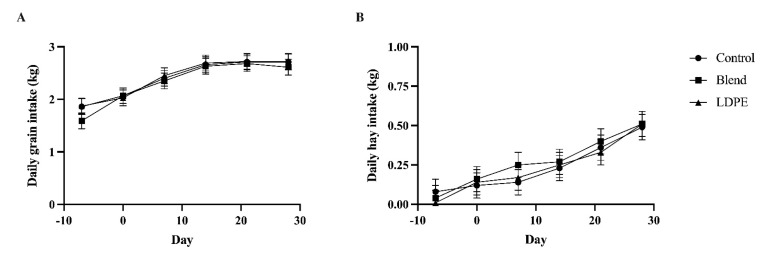
Average daily feed intake determined on a weekly basis, as fed, of Holstein bull calves given either empty gelatin capsules (Control), 13.6 g of a poly(butylene succinate-*co*-adipate) and polyhydroxyalkanoate melt-blend (Blend), or 13.6 g of low-density polyethylene (LDPE) for 30 d, separated by average daily grain intake (**A**) and average daily hay intake (**B**). For grain intake, Treatment: *p* = 0.35; Day: *p* < 0.01; Treatment x Day: *p* = 0.61. For hay intake, Treatment: *p* = 0.84; Day: *p* < 0.01; Treatment x Day: *p* = 0.58. Data are shown as least-squared means of treatments within time (LSM ± SEM).

**Figure 2 animals-13-00928-f002:**
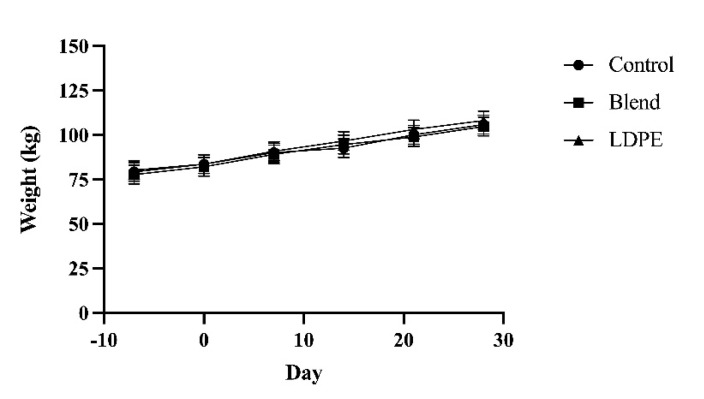
Weekly body weights of Holstein bull calves given Control, Blend, or LDPE treatments for 30 d. Treatment: *p* = 0.70. Day: *p* < 0.01. Treatment x Day: *p* = 0.15. Data are shown as least-squared means of treatments within time (LSM ± SEM).

**Figure 3 animals-13-00928-f003:**
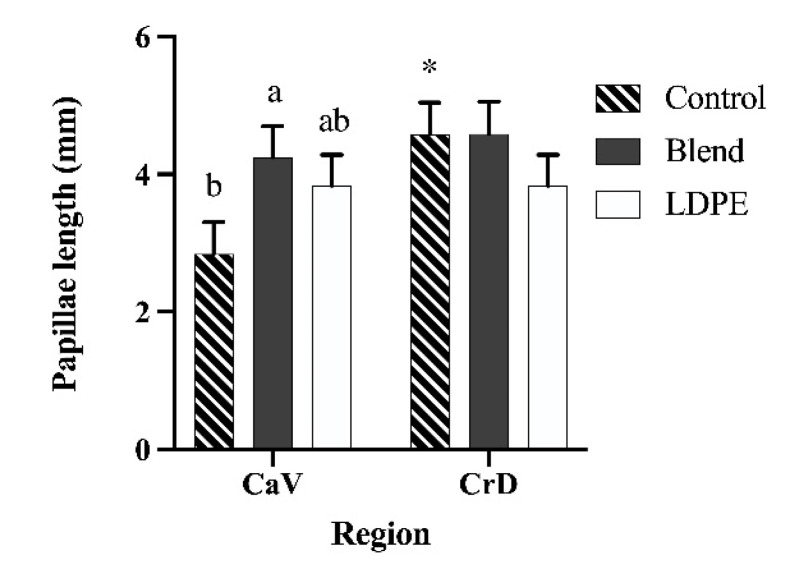
Papillae length in the caudal ventral (CaV) and cranial dorsal (CrD) regions of the reticulorumens of Holstein bull calves after 30 d of Control, Blend, or LDPE treatment administration. Treatment: *p* = 0.19. Region: *p* < 0.01. Treatment x Region: *p* < 0.01. ^a,b^ Different letters indicate differences between treatments within epithelium region. * Indicates a significant difference between regions within treatments. Data are shown as least-squared means of treatments within region (LSM ± SEM).

**Figure 4 animals-13-00928-f004:**
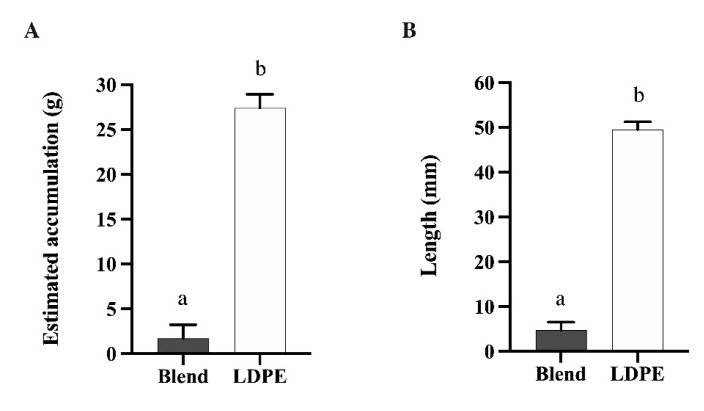
Characterization of polymer residues within rumen contents of Holstein bull calves given either Blend or LDPE treatment for 30 d, separated by (**A**) estimated total polymer accumulation and (**B**) average polymer length. For both parameters, treatment: *p* < 0.01. ^a,b^ Different letters indicate differences between treatments. Data are shown as least-squared means of treatments (LSM ± SEM).

**Table 1 animals-13-00928-t001:** Hematological parameters of Holstein bull calves at 0 d (Initial) and 30 d (Final) of administration of either 4 empty gelatin capsules (Control), 13.6 g of a poly(butylene succinate-*co*-adipate) and polyhydroxyalkanoate melt-blend in 4 gelatin capsules (Blend), or 13.6 g of low-density polyethylene in 4 gelatin capsules (LDPE).

Parameter	Control		Blend		LDPE		SEM	*p* <		
	Initial	Final	Initial	Final	Initial	Final		^1^ Trt	^2^ Ti	Trt × Ti
Red blood cell, cells/uL	10.63	10.33	11.57	11.24	10.54	10.71	0.50	0.25	0.66	0.80
Hemoglobin, g/dL	10.68	10.80	11.30	11.35	10.95	11.50	0.48	0.54	0.46	0.79
Hematocrit, %	36.03	35.80	38.03	37.23	36.65	38.18	1.48	0.47	0.89	0.72
Mean corpuscular volume, fL	33.88	34.78	32.90	33.10	34.83	35.65	0.68	0.02	0.06	0.62
Mean corpuscular hemoglobin concentration, g/dL	29.60	30.18	29.70	30.50	29.85	30.13	0.36	0.83	0.06	0.74
Red cell distribution width, %	27.63	26.18	27.95	27.75	26.88	26.33	1.16	0.72	0.03	0.26
Reticulocyte, cells/uL	0.85	1.75	1.15	1.98	1.80	1.63	0.58	0.81	0.16	0.40
White blood cell, cells/uL	9.67	8.13	9.19	8.43	10.17	7.55 ^x^	0.96	1.00	0.01	0.35
Neutrophil, cells/uL	4.48	2.55 ^x^	4.48	3.00	4.50	2.23 ^a^	0.70	0.82	<0.01	0.68
Lymphocyte, cells/uL	3.73	4.05	3.53	3.90	3.95	4.03	0.39	0.83	0.17	0.77
Monocyte, cells/uL	1.30	1.40	1.03	1.40	1.53	1.18	0.17	0.59	0.72	0.06
Eosinophil, cells/uL	0.00	0.03	0.05	0.03	0.08	0.00	0.03	0.69	0.23	0.15
Basophil, cells/uL	0.10	0.15	0.10	0.13	0.13	0.10	0.02	0.74	0.29	0.16
Platelets, cells/uL	684.75	526.00 ^a^	703.50	643.50	681.75	486.50 ^a^	33.56	0.10	<0.01	0.08
Mean platelet volume, fL	7.03	7.15	7.30	7.38	7.18	7.05	0.22	0.42	0.81	0.57
Erythrocyte sedimentation rate, mm/24 h	4.52	4.34	4.98	4.08	4.11	4.11	0.49	0.50	0.35	0.59

^1^ Treatment effect; ^2^ sampling time effect; ^a^ indicates a significant difference from the initial value within treatment, *p* < 0.05; ^x^ indicates a tendency to differ between time points within treatment, *p* < 0.10; data shown are the least-squared means.

**Table 2 animals-13-00928-t002:** Rumen parameters of Holstein bull calves after 30 d of administration of either Control, Blend, or LDPE treatment.

Parameter	Control	Blend	LDPE	SEM	*p* <
Forestomach, kg	16.04	14.57	16.19	1.29	0.24
Reticulorumen, kg	12.44	11.58	13.02	1.20	0.25
Empty reticulorumen, kg	2.11	2.08	2.20	0.18	0.63
Rumen contents, kg	10.33	9.50	10.82	1.09	0.35
Methylene blue reduction time, s	143 ^xy^	108 ^y^	154 ^x^	17	0.06
Rumen pH	6.8	6.8	6.9	0.1	0.61
Rumen temperature, °C	37.8	38.3	38.3	0.7	0.58

^x,y^ Different letter superscripts indicate tendencies for treatments to differ (*p <* 0.10); data shown are the least-squared means.

## Data Availability

There are no publicly available data due to commercial restrictions of the products.

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
