# Peer review of "Clearance of Biodegradable Polymer and Polyethylene Films from the Rumens of Holstein Bull Calves"

_animals, 2023, doi:10.3390/ani13050928_

Round 1

Reviewer 1 Report

The paper “Clearance of Biodegradable Polymer and Polyethylene Films 2 From the Rumens of Holstein Bull Calves” is aimed to detect the clearance of low-density polyethylene and a blend of PBSA:PHA from the rumen of veal calves together with the possible impact of the two treatment on animal health.

The study is well-designed and the text is very easy to read. The Material and Methods section gives sound information about the experimental design. The results are clearly described and the discussion provides a very complete explanation of the data obtained.

I only suggest the authors to better justify i) the choice of so young animals, and ii) the experimental time.

Reviewer 2 Report

The study addresses an important aspect of modern agriculture, which is plastic waste and the consequences of its ingestion by ruminants and other livestock animals.

 Major drawbacks I see in this study are the following:

1)     Lack of analysis of certain variables. Authors did not determine plastics in the feces. They often report the outwash from the rumen, but do not analyze the feces. It would have helped a lot to characterize & quantify plastic flows through the animal.

Also, no plastic residues or similar were analyzed in the blood, which would be interesting to see, where the biodegradable plastic goes. In my understanding, polyethylene is widely (not fully!) undegradable in the rumen and will be excreted if not retained in the rumen; therefore an environmental pollution risk. Whereas the biodegradable plastic may adsorbed in the intestine of the animal and potentially causing problems there; therefore a potentially bigger animal health risk at a systemic level.

Additionally, it would have been helpful if rumen pH and MBRT were determined at several time points (e.g. stomach tubing), and not only one snapshot event at the end of the trial. It cannot really be related to treatments by this.

2)     Confusion in the materials and methods. Authors have three treatments, but describe that calves were blocked in four blocks (L114) + in L133-134, authors again talk about four groups. Did they exclude a treatment? Is this only an excerpt of a larger trial?

Likewise, they state the trial took 30d, but report 6-week study in L112..

Also, I wonder if the calves were already completely weaned at d62. This sounds very early to me

3)     The results are poorly presented. In Table 1, authors use uppercase letters to illustrate significance, but in Table2 this is used for a tendency (btw, authors do not define significance and trends in the M&M, statistics). Likewise, the bar color assignments are not similar for treatments in Figures 3 and 4, which is confusing for the reader.

In Table 1, authors removed the P-value for interaction when not significant, but say they removed the interaction then from the model? Why would you do this? It will change the ANOVA and one does also not know if the P-values presented for main effects do now derive from the model including or excluding the interaction.

4)     Authors reveal in the study that animals were sick at the beginning of the experiment. They somehow qualify this statement by saying it would mainly apply to the acclimation period, but argument with the disease state in their discussion and admit that this should have affected the blood data, especially the immunological parameters. Thus, the findings have no significance – though I appreciate the authors to state openly that animals were sick.

5)     The discussion is too long and lacks focus on the key findings. The authors always mention that the study duration was too limited and the plastic particles too short, so why comparing it with so many other studies made at slaughterhouses. Also, I can imagine that ruminants sort out a lot of plastic under “normal feeding conditions” (at least we clearly see this at our experimental farm units with e.g. silo stretch foil) and those extreme results observed at slaughterhouses in Africa were rather caused by inferior feeding conditions.

+ What are the PHA- and PBSA-degrading microbes in the rumen that increase with maturation? Is there a reference for this statement? I can only think of a general increase in microbial degradation capacity as the rumen volume increases + a potential diet effect…

6)     The conclusions are a bit creative, so to say. In my opinion, one should not introduce new hypotheses (although this is generally a good thing for a scientist), because this is not the conclusion from the data. Also, statements on the marketing of new plastic nets have nothing to do with this study.

As a note, I think that ruminal plastic degradation is comparable to fiber degradation, meaning that not only the chemical structure of fiber/plastic determines the degradation and retention, but also the physical structure.

Reviewer 3 Report

The manuscript provides interesting information regarding the actual problem of ruminant plastic ingestion. Congratulation for your manuscript presented very clearly.

Attention to standardize the references.

Abstract

L25 material, while

Keywords:…; rumen, plastic impaction.

Introduction:

L53 specify better: 20% of cattle where???

Materials and methods

L90 A 10% PHA and 90% polymer blend (w/w)….

L133 not clear you speak about 2 groups start one day and the remaining 2 groups started the next, but the groups were 3 not 4. Please clarify it.

Results

L309 ….0.04), …

L334 verify character.

L353 health, while

References

Please standardize references as indicated in the “Instruction for the authors”.

  Journal Articles:
1. Author 1, A.B.; Author 2, C.D. Title of the article. Abbreviated Journal Name YearVolume, page range.

 Books and Book Chapters:
2. Author 1, A.; Author 2, B. Book Title, 3rd ed.; Publisher: Publisher Location, Country, Year; pp. 154–196.

Round 2

Reviewer 2 Report

Thank you for answering all my comments. I appreciate that authors took time to consider them. It is a nice manuscript.